# Vescalagin from Pink Wax Apple (*Syzygium samarangense* (Blume) Merrill and Perry) Protects Pancreatic β-Cells against Methylglyoxal-Induced Inflammation in Rats

**DOI:** 10.3390/plants10071448

**Published:** 2021-07-15

**Authors:** Wen-Chang Chang, James Swi-Bea Wu, Szu-Chuan Shen

**Affiliations:** 1Department of Food Science, National Chiayi University, Chiayi 600355, Taiwan; wcchang@mail.ncyu.edu.tw; 2Graduate Institute of Food Science and Technology, National Taiwan University, Taipei 10617, Taiwan; jsbwu@ntu.edu.tw; 3Graduate Program of Nutrition Science, National Taiwan Normal University, Taipei 10610, Taiwan

**Keywords:** vescalagin, methylglyoxal, inflammation, antioxidant, insulin secretion

## Abstract

Methylglyoxal (MG) is the primary precursor of advanced glycation end products involved in the pathogenesis of inflammation and diabetes. A previous study in our laboratory found anti-inflammatory and anti-hyperglycemic effects of the polyphenol vescalagin (VES) in rats with MG-induced carbohydrate metabolic disorder. The present study further investigated the occurrence of inflammation in pancreatic β-cells in MG-induced diabetic rats and the mechanism by which VES prevents it. The results showed that VES downregulates the protein expression levels of advanced glycation end product receptors and CCAAT/enhancer binding protein-β and upregulates the protein expression levels of pancreatic duodenal homeobox-1, nuclear factor erythroid 2-related factor 2 and glyoxalase I from the pancreatic cells. The results also revealed that VES elevates glutathione and antioxidant enzyme contents and then downregulates c-Jun N-terminal kinase and p38 mitogen-activated protein kinases pathways to protect pancreatic β-cells in MG-administered rats.

## 1. Introduction

Diabetes mellitus (DM) is a chronic disease associated with carbohydrate metabolism and caused by a deficiency in insulin secretion or by the ineffectiveness in insulin action [1]. The prevalence of DM, obesity, and many other metabolic syndromes has been linked to the increased consumption of methylglyoxal (MG)-containing foods [2,3]. MG is a major precursor of advanced glycation end products (AGEs) which in turn lead to oxidative stress [4]. Oxidative stress plays an important role in the pathophysiology of inflammation, insulin resistance, atherogenesis, and diabetes [5]. Cell studies indicated that MG and AGEs may promote the production of inflammatory cytokines that cause the damage of pancreatic β-cells [4,5,6]. MG has also been found from animal studies to cause inflammation in Sprague–Dawley rats, to induce their pancreatic impairment, and to affect insulin secretion as a consequence [5]. It appears that the protection of pancreatic β-cells can be an effective way for the maintenance of insulin secretion and the prevention of DM.

DM is associated with protein glycation [6]. AGEs may activate advanced glycation end product receptors (RAGE) on the membrane of pancreatic cells and then upregulate the two key inflammatory transcription factors, namely early growth response-1 and nuclear factor kappa B (NF-κB) [4,7]. NF-κB may promote the release of cytokines, including tumor necrosis factor alpha (TNF-α), interleukin-1 beta (IL-1β), and interleukin-6 (IL-6) [4]. High levels of these cytokines and reactive oxygen species (ROS) reduce the release of translocation-specific transcription factor pancreatic duodenal homeobox-1 (PDX-1) from the nucleus of a β-cell to the cytoplasm and hinder the synthesis of insulin [8]. Exposure to MG may decrease insulin secretion and the survival rate of pancreatic β-cells via redox-independent inhibition of phosphatidylinositol 3-kinase insulin signal pathway [9]. There are positive correlations between MG content and the survival rates of pancreatic β-cells both in vivo and in vitro [5,9].

Serum MG levels are less than 1 μM in healthy humans but can be elevated to 2–6 μM in diabetic patients, with a positive correlation with the degree of hyperglycemia [10]. Recently, many foods, including steak, wine, and beer, were found to be associated with high MG levels in the serum of human blood. Broiled steak has been reported to contain 12.73 μg/g of MG in a study in the United States [11] Certain samples of wine and beer were found to contain 21.59 and 13.88 μM MG, respectively, in Europe [12,13]. Under physiological conditions, glutathione (GSH) may activate glyoxalase to convert MG into D-lactate [6]. Glyoxalase I (GLO-1) and glyoxalase II (GLO-2) may also catalyze the detoxifying conversion of MG into D-lactoylglutathione and D-lactate, suggesting that GLO-1 may retard MG-induced formation of AGEs [14]. Nuclear factor erythroid 2-related factor 2 (Nrf2) has been reported to promote the expression of GLO-1 and the conversion of MG to D-lactate [15]. Many antioxidants, including quercetin and phenolic acids, have been found to attenuate oxidative damage by activating Nrf2 [16].

Ellagitannins are bioactive polyphenols with antioxidant and anti-inflammatory activities [17]. Pink wax apple (*Syzygium samarangense* (Blume) Merrill and Perry cv. Pink) fruit contains the ellagitannin vescalagin (VES). VES has been reported to be antitumor, cardiovascular disease preventive, insulin-resistance alleviative, and dyslipidemia mediative [18,19]. In a previous study, our laboratory found that VES may reduce serum glucose content and in the meanwhile increase serum insulin and C-peptide levels [20]. No studies with regard to the protective effect of VES on pancreatic β-cells have been reported yet. The aim of the present study was to elucidate the mechanism by which VES ameliorates the MG-induced inflammation and insulin secretion reduction by assessing the activities of oxidation enzymes, inflammatory proteins, and the insulin secretion-related proteins in pancreatic β-cells in rats.

## 2. Results

### 2.1. Diet Intake and Body Weight in Rats Orally Administered with MG

Table 1 shows the diet intake and body weight in rats orally administered with MG. In the present study, the rats administered orally once a day with MG at 300 mg/kg followed with another chemical (PIO, AG, or VES) at 30 mg/kg for a period of 8 weeks showed no significant difference in diet intake, drink intake, or body weight as compared with the normal group (Table 1), indicating no occurrence of acute-phase response in rats in the feeding period.

### 2.2. Amylase and Lipase Activities in the Rats

Figure 1 shows the effect of VES on amylase and lipase activities in MG-administered rats. When the contents of amylase and lipase are increased in the blood, this is recognized as a phenomenon of pancreatic cell inflammation in mammals. However, if the contents of amylase and lipase are significantly increased, it is recognized as acute pancreatitis. There were no significant differences in serum lipase activity among all groups. MG group (2259 ± 124 U/L) showed higher serum amylase activity as compared with the normal group (1869 ± 142 U/L) (*p* < 0.05), indicating that MG may induce hyperglycemia in rats. Our results show significantly lower activities of serum amylase in MG+PIO, MG+AG, and MG+VES treatment groups, by 19%, 35.8%, and 27.6%, respectively, as compared with the MG group (*p* < 0.05) (Figure 1). We proposed that VES may ameliorate hyperglycemia in MG-administered rats via the inhibition of carbohydrate digestion enzymes.

### 2.3. RAGE and Insulin Secretion-Related Protein Expression Levels in Pancreatic β-Cells of the Rats

Figure 2 shows the effect of VES on the expression levels of RAGE and insulin secretion-related proteins in pancreatic β-cells of MG-administered rats. The results indicate that the expression of RAGE and C/EBPβ proteins in pancreatic β-cells can be elevated by MG and the elevation can be offset by PIO, AG, and VES (*p* < 0.05) (Figure 2A,B). The expression levels of pancreatic PDX-1 and GLO-1 proteins were increased in rats administered with MG and PIO, AG, or VES (Figure 2C,D). An increase in PDX-1 expression and a decrease in C/EBPβ expression correspond to an increase in insulin secretion and glucose tolerance [5]. We speculate that VES inhibits AGE formation and inflammation reaction in pancreatic cells. The present study also found significant reductions in pancreatic GSH content in the MG group as compared with normal group (*p* < 0.05). Furthermore, those groups with PIO, AG, and VES in the diet showed 202%, 154%, and 311% elevations in pancreatic GSH content, respectively, in comparison with the MG group (*p* < 0.05) (Figure 2E). The results suggest that VES elevates GSH content to protects pancreatic cells in MG-administered rats.

### 2.4. Expression Levels of Inflammation Proteins in Pancreatic β-Cells of the Rats

Figure 3 shows the effects of VES on the expression levels of inflammation proteins in pancreatic β-cells of MG-administered rats. MG and AGEs can increase oxidative stress and promote the generation of inflammatory cytokines, as found in our previous study [20]. The present study shows that pancreatic levels of the inflammatory proteins NF-κB, ICAM-1, and TNF-α were increased in MG-administered rats (*p* < 0.05). The pancreatic levels of NF-κB, ICAM-1, and TNF-α protein were reduced in MG-administered rats by feeding with VES (Figure 3A,B,D). It was found that PIO, AG, and VES activated pancreatic Nrf2 in MG-administered rats (Figure 3E). Many antioxidants, such as VES and PIO, have been evaluated for their ability to activate Nrf2 and to attenuate oxidative damage and inflammatory reactions [6].

### 2.5. Activities of Antioxidant Enzymes in Pancreatic β-Cells of the Rats

Figure 4 shows the effect of VES on antioxidative enzymes in pancreatic β-cells of MG-administered rats. The present study showed that MG downregulated the expression levels of Nrf2 and antioxidant enzymes, including GSH, SOD, and catalase, indicating the occurrence of redox imbalance after the administration of MG (Figure 2E and Figure 4A,B). The present study also found that GSH and catalase contents were elevated in pancreatic β-cells in MG+PIO, MG+AG, and MG+VES groups as compared with the MG group (Figure 2E and Figure 4B). In comparison with the MG group, those groups administered with MG and PIO, AG, or VES in the diet showed significant reduction of 47%, 52%, or 57%, respectively, in pancreatic MDA content (*p* < 0.05) (Figure 4C). The present study showed that activation of Nrf2 in pancreatic β-cells reduces inflammatory reaction, which may ameliorate β-cell damage (Figure 3). We speculated that PIO, AG, and VES could effectively activate Nrf2 by phosphorylation, thereby increasing antioxidant enzyme contents and suppressing lipid peroxidation in the pancreas of the MG-administered rat.

### 2.6. Mitogen-Activated Protein Kinases (MAPKs) and Inflammation in Pancreatic β-Cells of the Rats

Finally, according to the above observations, we will confirm whether VES can ameliorate serum glucose by downregulating the inflammatory proteins of the common inflammation pathway from pancreatic cells. Figure 5 shows the effect of VES on the expression levels of MARKs in pancreatic β-cells of MG-administered rats. The pancreatic levels of JNK and p38 phosphorated proteins were reduced in rats administered with VES in addition to MG, suggesting that VES may downregulate the phosphorylation of JNK and p38 proteins in MAPK pathways and protect pancreatic β-cells in MG-administered rats.

## 3. Discussion

Recent studies showed that MG injection of Sprague–Dawley rats may cause glucose intolerance, reduce the insulin-stimulated glucose uptake in adipose tissue, and result in pancreatic dysfunction and that oral administration of MG at 300 mg/kg/day may induce inflammation in the pancreatic β-cells [5,10,20]. These animal studies also showed that there was no significant difference in diet intake, drink intake, and body weight among all groups. The aforementioned studies are consistent with our results.

Some chemicals, such as MG and AGEs, may damage the pancreatic tissue and even lead to pancreatitis and diabetes [2,5,21]. Recent studies demonstrated that amylase and lipase may enhance carbohydrate and lipid digestion in the induction of hyperglycemia and hyperlipidemia in rats [22,23]. Mnafgui et al. found elevated amylase activities in the serum and pancreas of diabetic rats [24]. The increase in α-amylase activity also occurred in the rats orally administered with MG, indicating that oral administration is a feasible method to induce hyperglycemia in the animal [20]. The elevated serum amylase activity may be associated with pancreatic β-cell inflammation in rats [24]. A range of berry polyphenols, for example, flavonols, anthocyanidins, resveratrol, ellagitannins, and proanthocyanidins, can inhibit amylase to a level that affects starch degradation in rats [25]. Liu et al. proposed that polyphenols and flavonoids act as antioxidants to suppress the activity of amylase in vitro [23]. Our results show a significantly lower activity of serum amylase in the MG+VES treatment group as compared with the MG group (*p* < 0.05) (Figure 1). We reported that the effect of MG in reducing the insulin and C-peptide contents in serum, upregulating inflammatory cytokines, and inducing hyperglycemia in MG-administered rats previously [20]. Both amylase and lipase levels have been used as indices for the inflammation of the pancreas in rats [21]. Previous studies demonstrated that VES could trap MGO through a conjugating reaction and thus inhibit further glycation reaction. VES was able to inhibit both α-amylase and α-glucosidase [20,25]. We speculate that MG promotes AGEs production and inflammation and hurts pancreatic β-cells in rats. The following assessments with regards to the effects of VES on inflammation and the insulin secretion of pancreatic β-cells in rats administered with MG were performed to support the postulation.

RAGE downregulates the expression levels of insulin secretion-related proteins and induces inflammation in pancreatic β-cells [4,6]. Pioglitazone is an insulin sensitizer used for controlling the serum glucose of type 2 diabetes mellitus. The antidiabetic drug PIO is a peroxisome proliferator-activated receptor-γ (PPARγ) ligand. The activation of PPARγ is known to affect pancreatic β-cell functions, including insulin production [26]. AG is a nucleophilic hydrazine derivative and is well known for its action in blocking advanced glycation, and it has been shown to prevent diabetic complications. AG was reported to block glycation, and it also inhibits the production of toxic aldehydes by quinone enzymes [27]. AG, PIO, and VES are effective agents in scavenging MG molecules, downregulating RAGE expression, and retarding AGE formation [20,28]. Reduced MG intake may alleviate the production of the pancreatic serum proteins that are promotive for RAGE expression, glycation, and inflammation [2,4,29]. The results in the present study reconfirm the above-mentioned findings. RAGE may downregulate PDX-1 protein expression in pancreatic RIN-m5F cells [16]. In β-cells, PDX-1 protein is a positive regulator of insulin synthesis, whereas C/EBPβ protein is a repressor of insulin gene transcription [5]. MG was found to downregulate PDX-1, to upregulate C/EBPβ, and to inhibit insulin secretion from isolated β-cells from rats [5]. The expression of pancreatic PDX-1 protein was increased in rats administered with MG and PIO, AG, or VES. The expression of pancreatic C/EBPβ protein was increased by the administration with MG, and the increment could be offset by PIO, AG, and VES (Figure 2). The present study indicates that VES is promotive of insulin secretion and glucose tolerance in rats.

Dhar et al. proposed that the reduction in MG content may ameliorate insulin resistance and β-cell damage [5]. Reflecting the impact of MG on biological systems, cells developed a GLO enzyme system that is dedicated to MG degradation and was well conserved during evolution [4]. The GLO system is present in the cytosol of all animal cells and comprises two enzymes, GLO-1 and GLO-2, and a catalytic amount of GSH [30]. GLO-1 and GLO-2 catalyze MG to D-lactoylglutathione and D-lactate and thereby suppress dicarbonyl-mediated glycation reactions [30]. The present study found that the pancreatic GLO-1 protein expression was elevated in MG-administered rats administered with PIO, AG, or VES (*p* < 0.05) (Figure 2D), suggesting that the increase in GLO-1 expression is involved in the mechanism for MG reduction. A high expression of GLO-1 prevented the increase of MG and AGEs in streptozotocin-induced diabetic rats [31]. The activity of the GLO enzyme system is vital to mitigate the dicarbonyl components, such as MG and AGEs, oxidative stress, and pancreatic β-cell damage [30]. In a previous study, we found the effectiveness of VES in ameliorating the serum glucose level in the rat via the reduction in MG and AGE contents [20]. GLO-1 activity is GSH-dependent [4,16,32]. We propose that VES elevates GSH content (Figure 2D), promotes GLO-1 expression (Figure 2E) in MG-administered rats, accelerates metabolism of MG, attenuates AGE formation in pancreatic β-cells, and thereby protects these cells.

The elevation in MG and inflammation cytokines may induce abnormality in β-cells [20]. The molecular mechanism underlying MG toxicity appears to be complex and may involve the glycation of cellular proteins and induction of cell death [4,16]. MG and AGEs promote the release of proinflammatory cytokines, increase the activity of many pro-oxidant enzymes such as NADPH oxidase and JNK, and upregulate the expression of NF-κB in pancreatic cells in rats [4,20]. The contents of inflammatory cytokines TNF-α and IL-6 are elevated in the serum of MG-administered rats [20]. The present study shows the pancreatic levels of inflammatory proteins were reduced in MG-administered rats by feeding with VES (Figure 3), suggesting that VES may reduce the expression levels of inflammation protein to protect pancreatic β-cells. Previous studies showed that antioxidants, including PIO and VES, may reduce serum contents of TNF-α and IL-6 in MG-administered rats [6,16,20]. We found that VES activated pancreatic Nrf2 in MG-administered rats (Figure 3E). Nrf2 has been shown to improve glucose tolerance, insulin sensitivity and metabolic syndrome in rats and to inhibit oxidative stress and inflammatory reactions in the pancreas [33,34,35]. We propose that VES downregulates the expression levels of inflammation proteins, elevates Nrf2 expression, and protects β-cells in MG-administered rats (Figure 3).

Cytokine-induced β-cell dysfunction can be reduced through inhibition of ROS production, alleviation of inflammatory reaction, or increase in the activities of antioxidative enzymes [8]. Nrf2 is a crucial regulator of the cellular redox homeostasis and plays a pivotal role, through the promotion of the expression levels of antioxidant enzymes and other antioxidant proteins, in protecting cells against ROS damage and inflammation [36]. The antioxidant enzymes controlled by Nrf2 include catalase, glutathione peroxidase, GSH, and superoxide dismutase (SOD) [36]. Previous studies showed that PPARγ agonists, such as PIO, could exhibit a regulatory effect upon Nrf2 [16,36]. Previous studies demonstrated that MG promotes the generation of MDA, which is a product of lipid peroxidation and a marker of dysfunctional insulin expression and diabetes [6,37]. GSH is well known to serve diverse biological functions, including the reduction in MDA content, the alleviation of inflammation, and the protection of pancreatic β-cells [37,38]. The present study suggested that PIO, AG, and VES elevated GSH content in MG-administered rats via the activation of Nrf2 protein expression (Figure 2E and Figure 3E).

Studies indicated that MAPKs play a pivotal role in the development of insulin resistance induced by various factors, including inflammatory cytokines, ROS, chemicals, and oxidants [4,36,39]. MG was observed to activate MAPK pathways of inflammation in osteoblasts, endothelial cells, and Jurkat leukemia cells in a redox-dependent manner [4]. The present study showed that the pancreatic levels of JNK and p38 phosphorated proteins were reduced in rats administered with VES in addition to MG (Figure 5), suggesting that VES may downregulate the phosphorylation of JNK and p38 proteins in MAPK pathways and protect pancreatic β-cells in MG-administered rats.

MG may promote the phosphorylation of p38 in pancreatic islets via the elevation in expression levels of proinflammation proteins such as NF-κB [8]. Reduction in the MG content often alleviates inflammation [4,6,10,14]. In a previous study we found VES to reduce the MG content in the serum of MG-administered rats [20]. VES is a polyphenol with antioxidant and anti-inflammatory bioactivities [17]. The present study showed that VES promotes the activities of antioxidant enzymes, downregulates the expression of inflammation factor NF-κB and the phosphorylation and expression levels of inflammation proteins JNK and p38 MAPKs in pancreatic β-cells (Figure 3 and Figure 4), and increases insulin secretion in MG-administered rats (Figure 5). Based on the above-described results, we propose that VES downregulates the expression levels of inflammation proteins and protects the pancreatic β-cells in MG-administered rats via the inactivation of JNK and p38 MAPK pathways.

## 4. Materials and Methods

### 4.1. Chemicals

VES was extracted and purified from unripe pink wax apple fruit (*Syzygium samarangense* (Blume) Merrill and Perry) following the reported procedure [40]. 1,4-Dithiothreit, acrylamide, aminoguanidine (AG), ammonium peroxydisulfate, D-glucose, ethyl alcohol, ethyl ether, glycine, methanol, MG, pioglitazone hydrochloride (PIO), sodium chloride, sodium dodecyl sulfate (SDS), sodium phosphate dibasic, sulfuric acid, thiourea, tris base, Triton X-100, Tween-20, and urea were purchased from Sigma (St. Louis, MO, USA).

### 4.2. Animals and Diets

Male Wistar rats (5-week old) were supplied by National Laboratory Animal Breeding and Research Center, Taipei, Taiwan. The rats were maintained in standard laboratory conditions at 22 ± 1 °C on 12-h light/12-h dark cycle with free access to food and water for the duration of the study. The room conditions and treatment procedures were in accordance with the National Institutes of Health (NIH) Guide for the Care and Use of Laboratory Animals, and all protocols were approved by the Institutional Animal Care and Use Committee of National Taiwan Normal University, Taipei, Taiwan (approval number 103042). The rats were fed a normal diet and deionized water for 1 week and then divided into 5 groups, 8 animals each, to feed for 8 more weeks. Among them, 1 group was fed a normal diet and 30 mg/mL deionized water (Normal group), while the other 4 groups were fed with the normal diet supplemented with MG at 300 mg/kg b.w./day and orally administered, once a day, with 30 mg/mL deionized water (MG group) or the solution of PIO (MG+PIO group), AG (MG+AG group), or VES (MG+VES group) at 30 mg/kg body weight/day. The dosage of MG followed the experimental designs reported by Dhar et al. and Lee et al. [5,6]. All animals were sacrificed by ethyl ether asphyxia at the end of the feeding period. The following operations were then performed.

### 4.3. Blood and Pancrease Tissue Sample Collection

Blood samples were taken from venter vein of the sacrificed rat, allowed to clot for 30 min at room temperature, and then centrifuged at 3000× *g* for 20 min to obtain the serum, which was stored at −80 °C before use. The pancreas tissue was picked out and rinsed with cold modified Tyrode calcium-free solution. The tissue was cut into small pieces (2–3 mm) in Tyrode buffer and washed several times with this solution to remove blood and possible fat tissue contamination and stored at −80 °C before use.

### 4.4. Biochemical Analyses

ELISA kits for rat α-amylase, catalase, GSH, lipase, superoxide dismutase (SOD), and thiobarbituric acid reactive substances (TBARS) were purchased from Randox Laboratories (Crumlin, Antrim, UK). Analyses were performed following the supplier’s protocols. Briefly, pancreas tissues were homogenized on ice with RIPA buffer (Cell Signaling Technology, Beverly, MA, USA) using a homogenizer and centrifuged at 12,000× *g* (4 °C, 60 min) to collect supernatant as the pancreas extract. The protein concentration in the extract was determined using Bio-Rad Protein Assay Kit (Bio-Rad, Hercules, CA, USA) with bovine serum albumin as the standard. Pancreas extract was mixed with the GSH commercial reaction reagent, and the absorbance at OD450 nm was measured in kinetic mode for 40–60 min at room temperature by ELISA reader. Data were further analyzed following the GSH kit formula. Pancreas extract was mixed with the SOD commercial reaction reagent (WST-1 and enzyme working solution) and then incubated at 37 °C for 20 min. Absorbance was then measured at 450 nm with a spectrophotometer at room temperature. Data were analyzed following the SOD kit formula. Pancreas extract was mixed the catalase commercial reaction reagent, and then the supplier’s protocols were followed (1. add H_2_O_2_ to samples and incubate 30 min at 25 °C; 2. add stop solution to samples; 3. add development mix and incubate 10 min at 25 °C). Absorbance was then measured at 570 nm with a spectrophotometer. Data were analyzed following the catalase kit formula. Malondialdehyde (MDA), other aldehydes, and lipid hydroperoxides are able to form adducts with TBA. The MDA concentration was used as an index of lipid peroxidation using the thiobarbituric acid reactive substances (TBARS) method. Pancreas extract was added to a mixture containing 0.5% TBA and 3.75% BHT in methanol. Extracts were heated in a boiling water bath for 30 min and then cooled on ice. The MDA–TBA adduct can be easily quantified colorimetrically at 532 nm. Data were analyzed following the MDA kit formula.

### 4.5. Western Blot Analysis

Aliquots of the pancreas extract, containing 100 μg protein in each, were used to evaluate the expression levels of CCAAT/enhancer binding protein-β (C/EBPβ), extracellular signal-regulated kinase (ERK), phosphorylated ERK (p-ERK), GLO-1, intercellular adhesion molecule 1 (ICAM-1), c-Jun N-terminal kinase (JNK), phosphorylated p-JNK (JNK), monocyte chemoattractant protein-1 (MCP-1), NF-κB, Nrf2, phosphorylated Nrf2 (p-Nrf2), p38 MAPK (p38), phosphorylated p38 (p-p38), PDX-1, RAGE, and TNF-α. The protein extract was separated by using dodecyl sulfate polyacrylamide gel electrophoresis on 10% polyacrylamide gel and transferred to a polyvinylidene difluoride membrane (Millipore, Billerica, MA, USA). The membrane was incubated with block buffer (PBS containing 0.05% Tween-20 and 5% *w*/*v* nonfat dry milk) for 1 h and washed with PBS containing 0.05% Tween-20 (PBST) 3 times. Then, it was probed with each of the 1:1000 diluted solutions of anti-NF-κB, anti-MCP-1, and anti-ICAM-1 (GeneTex, Irvine, CA, USA); 1:1000 diluted solutions of anti-ERK, anti-p-ERK, anti-Nrf2, anti-PDX-1, anti-RAGE, and anti-TNF-α (Cell Signaling Technology, Beverly, MA, USA); and 1:1000 diluted solutions of anti-C/EBPβ, anti-GLO-1, anti-JNK, anti-p-JNK, anti-p-Nrf2, anti-p38, and anti-p-p38 (Epitomics, Burlingame, CA, USA) overnight at 4 °C. The intensity of the blots probed with 1:2000 diluted solution of rabbit monoclonal antibody to bind actin (Gene Tex, Irvine, CA, USA) was used as the control to ensure that a constant amount of protein was loaded into each lane of the gel. The membrane was washed with PBST 3 times, 5 min each; shaken in a solution of horseradish peroxidase-linked anti-mouse IgG or anti-rabbit IgG secondary antibody (Gene Tex, Irvine, CA, USA); washed with PBST 3 more times, 5 min each; and then exposed to the enhanced chemiluminescence reagent (Millipore, Bedford, MA, USA) following the manufacturer’s instructions. The films were scanned and analyzed using the UVP Biospectrum image system (Level, Cambridge, UK).

### 4.6. Statistical Analysis

Results expressed as means ± SD were analyzed by one-way ANOVA and Duncan’s new multiple range tests. All the *p* values less than 0.05 were considered to be significant.

## 5. Conclusions

The present study elucidates the mechanism by which VES alleviates MG-caused inflammation in pancreatic β-cells by evaluating the activities of antioxidant enzymes and the expression levels of anti-inflammation proteins in rats. Our experiment results revealed that VES elevates GSH content; upregulates the protein expression levels of Nrf2, GLO-1, and antioxidant enzymes; downregulates JNK and p38 MAPK pathways; and thereby protects pancreatic β-cells and improves insulin secretion in MG-administered rats. These findings support the potential for VES to become a health food ingredient in the prevention of diabetes.

## Figures and Tables

**Figure 1 plants-10-01448-f001:**
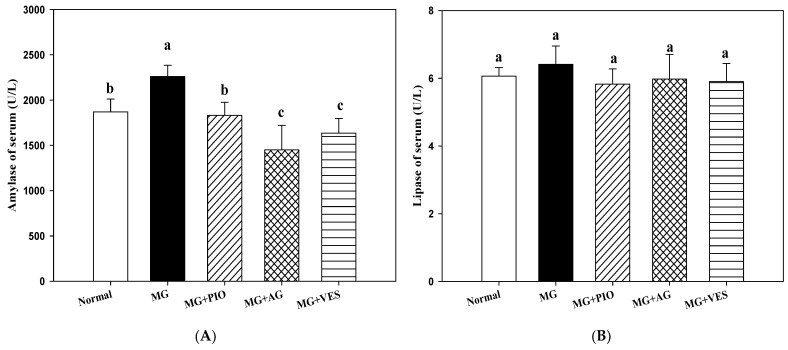
Effects of pioglitazone, aminoguanidine, and vescalagin on pancreatic inflammation indices (**A**) amylase concentration and (**B**) lipase concentration in the serum of MG-administered rats. Normal: rats fed with normal diet and deionized water. MG: rats fed with normal diet with the addition of methylglyoxal (300 mg/kg body weight/day). MG+PIO: rats fed with normal diet with the addition of methylglyoxal (300 mg/kg body weight/day) and pioglitazone (30 mg/kg body weight/day). MG+AG: rats fed with normal diet with the addition of methylglyoxal (300 mg/kg body weight/day) and aminoguanidine (30 mg/kg body weight/day). MG+VES: rats fed with normal diet with the addition of methylglyoxal (300 mg/kg body weight/day) and vescalagin (30 mg/kg body weight/day). Feeding period was 8 weeks. Different letters (a–c) signify a statistically significant difference at *p* < 0.05. Results are from 8 repetitions and expressed as mean ± SD.

**Figure 2 plants-10-01448-f002:**
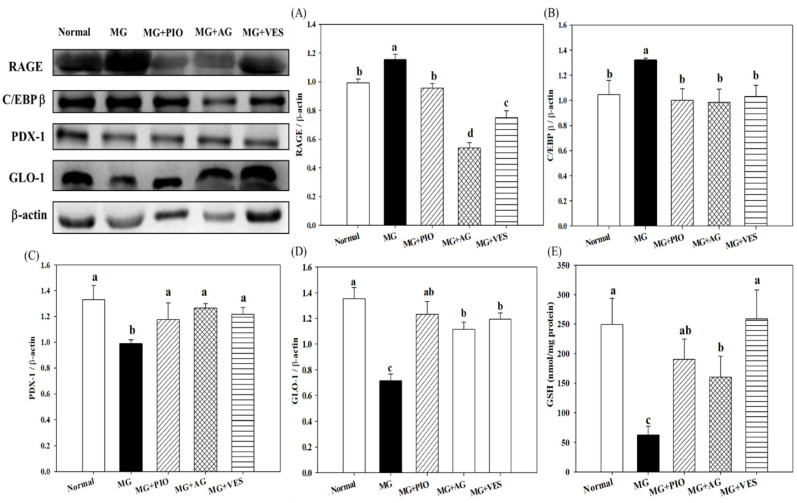
Effects of pioglitazone, aminoguanidine, and vescalagin on (**A**) RAGE expression, (**B**) C/EBP β expression, (**C**) PDX-1 expression, (**D**) GLO-1 expression, and (**E**) GSH concentration in the pancreatic β-cells of MG-administered rats. Normal: rats fed with normal diet and deionized water. MG: rats fed with normal diet with the addition of methylglyoxal (300 mg/kg body weight/day). MG+PIO: rats fed with normal diet with the addition of methylglyoxal (300 mg/kg body weight/day) and pioglitazone (30 mg/kg body weight/day). MG+AG: rats fed with normal diet with the addition of methylglyoxal (300 mg/kg body weight/day) and aminoguanidine (30 mg/kg body weight/day). MG+VES: rats fed with normal diet with the addition of methylglyoxal (300 mg/kg body weight/day) and vescalagin (30 mg/kg body weight/day). Feeding period was 8 weeks. Different letters (a–c) signify a statistically significant difference at *p* < 0.05. Results are from 8 repetitions and expressed as mean ± SD.

**Figure 3 plants-10-01448-f003:**
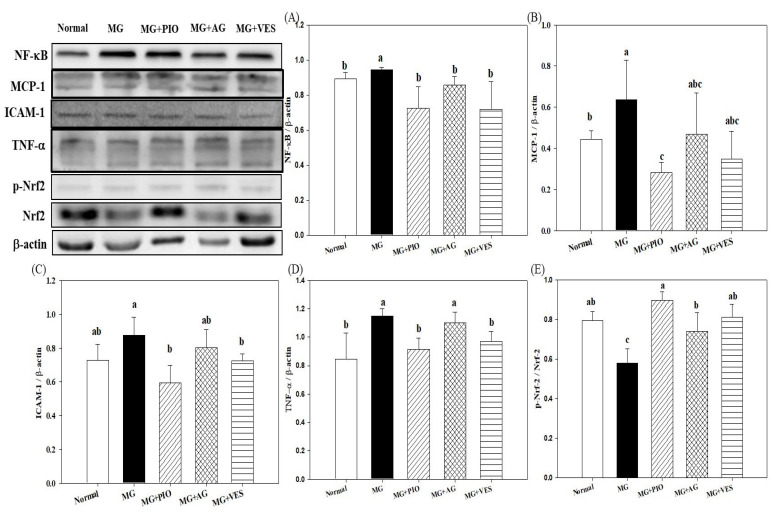
Effects of pioglitazone, aminoguanidine, and vescalagin on the expression levels of inflammation proteins (**A**) NF-κB, (**B**) MCP-1, (**C**) ICAM-1, (**D**) TNF-α, and (**E**) Nrf2 in the pancreatic β-cells of MG-administered rats. MG: rats fed with normal diet with the addition of methylglyoxal (300 mg/kg body weight/day). MG+PIO: rats fed with normal diet with the addition of methylglyoxal (300 mg/kg body weight/day) and pioglitazone (30 mg/kg body weight/day). MG+AG: rats fed with normal diet with the addition of methylglyoxal (300 mg/kg body weight/day) and aminoguanidine (30 mg/kg body weight/day). MG+VES: rats fed with normal diet with the addition of methylglyoxal (300 mg/kg body weight/day) and vescalagin (30 mg/kg body weight/day). Feeding period was 8 weeks. Different letters (a–c) signify a statistically significant difference at *p* < 0.05. Results are from 8 repetitions and expressed as mean ± SD.

**Figure 4 plants-10-01448-f004:**
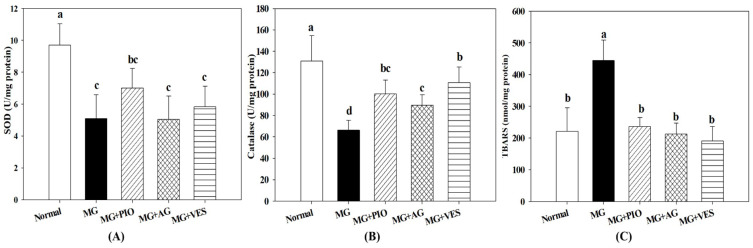
Effects of pioglitazone, aminoguanidine, and vescalagin on the concentrations of antioxidant indicators (**A**) SOD, (**B**) catalase, and (**C**) TBARS in the pancreatic β-cells of MG-administered rats. MG: rats fed with normal diet with the addition of methylglyoxal (300 mg/kg body weight/day). MG+PIO: rats fed with normal diet with the addition of methylglyoxal (300 mg/kg body weight/day) and pioglitazone (30 mg/kg body weight/day). MG+AG: rats fed with normal diet with the addition of methylglyoxal (300 mg/kg body weight/day) and aminoguanidine (30 mg/kg body weight/day). MG+VES: rats fed with normal diet with the addition of methylglyoxal (300 mg/kg body weight/day) and vescalagin (30 mg/kg body weight/day). Feeding period was 8 weeks. Different letters (a–c) signify a statistically significant difference at *p* < 0.05. Results are from 8 repetitions and expressed as mean ± SD.

**Figure 5 plants-10-01448-f005:**
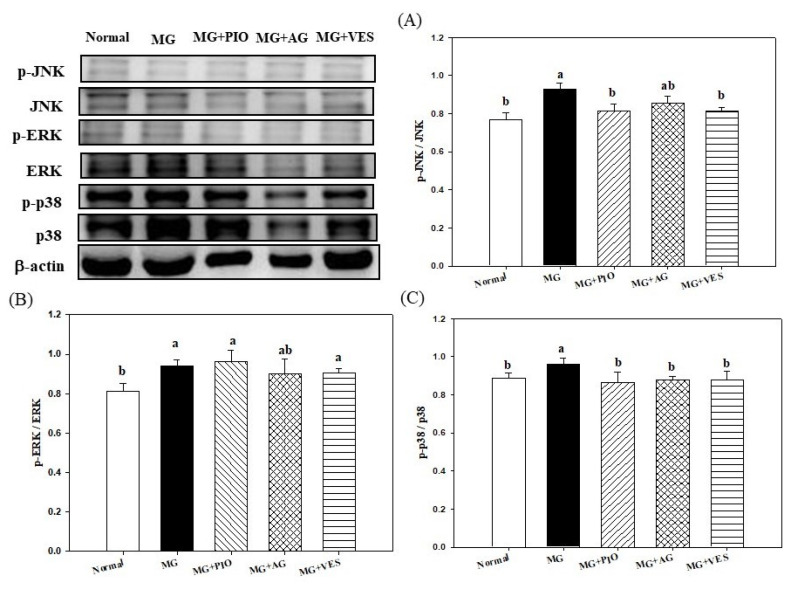
Effects of pioglitazone, aminoguanidine, and vescalagin on the expression levels of inflammation proteins (**A**) JNK, (**B**) ERK, and (**C**) p38 in the pancreatic β-cells of MG-administered rats. MG: rats fed with normal diet with the addition of methylglyoxal (300 mg/kg body weight/day). MG+PIO: rats fed with normal diet with the addition of methylglyoxal (300 mg/kg body weight/day) and pioglitazone (30 mg/kg body weight/day). MG+AG: rats fed with normal diet with the addition of methylglyoxal (300 mg/kg body weight/day) and aminoguanidine (30 mg/kg body weight/day). MG+VES: rats fed with normal diet with the addition of methylglyoxal (300 mg/kg body weight/day) and vescalagin (30 mg/kg body weight/day). Feeding period was 8 weeks. Different letters (a, b) signify a statistically significant difference at *p* < 0.05. Results are from 8 repetitions and expressed as mean ± SD.

**Table 1 plants-10-01448-t001:** Diet intakes, drink intakes, and body weights of rats administered with methylglyoxal and pioglitazone, aminoguanidine, or vescalagin.

Items/Groups	Normal	MG	MG+PIO	MG+AG	MG+VES
Diet intake (g/rat/day)	28.93 ± 5.66 ^a^	29.60 ± 5.42 ^a^	30.48 ± 5.41 ^a^	27.60 ± 5.76 ^a^	28.99 ± 4.84 ^a^
Drink intake (mL/rat/day)	42.40 ± 3.29 ^a^	44.23 ± 3.67 ^a^	49.67 ± 5.09 ^a^	43.42 ± 4.76 ^a^	46.63 ± 4.78 ^a^
Body weight (g)	416.11 ± 9.58 ^a^	444.50 ± 19.0 ^a^	420.64 ± 17.6 ^a^	411.01 ± 36.3 ^a^	424.37 ± 21.6 ^a^

Normal: rats fed with normal diet and deionized water. MG: rats fed with normal diet with the addition of methylglyoxal (300 mg/kg body weight/day). MG+PIO: rats fed with normal diet with the addition of methylglyoxal (300 mg/kg body weight/day) and pioglitazone (30 mg/kg body weight/day). MG+AG: rats fed with normal diet with the addition of methylglyoxal (300 mg/kg body weight/day) and aminoguanidine (30 mg/kg body weight/day). MG+VES: rats fed with normal diet with the addition of methylglyoxal (300 mg/kg body weight/day) and vescalagin (30 mg/kg body weight/day). Feeding period was 8 weeks. Different letters (^a^) on the same line signify a statistically significant difference at *p* < 0.05. Results are from 8 repetitions and expressed as mean ± SD.

## Data Availability

Data available in a publicly accessible repository.

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
