# Peer review of "Vescalagin from Pink Wax Apple (Syzygium samarangense (Blume) Merrill and Perry) Protects Pancreatic β-Cells against Methylglyoxal-Induced Inflammation in Rats"

_plants, 2021, doi:10.3390/plants10071448_

Round 1
Reviewer 1 Report
Publish with major revision
In general, this study learnt the effects of polyphenol vescalagin (VES) on the MG-induced inflammation and insulin secretion. The manuscript is recommended for publication with major revisions and the details are list below.
Major changes
- There is no introduction to the purpose of using pioglitazone and aminoguanidine in this study. Why are these selected for the study besides the VES?
- In the results section, the level of several groups of targets are learnt, including 1. Amylase and lipase; 2. RAGE and insulin secretion-related protein; 3. Inflammation proteins; 4. Antioxidant enzymes; and 5. Kinases. However, it was not explained in the study introduction why some of these proteins are selected, such as the selection of kinases. Please provide a more systematic introduction on how all of these types of proteins are selected as readout to evaluate the effect of VES.
- In all the figures, please describe the type of experiment conducted to obtain the corresponding figure. For example, what is the approach used to get figure 1? What is the readout of the experiment?
- For all the figures, what are the corresponding p-values of different letters a-c?
- Please provide the experimental methods for collecting pancreatic beta cells for ELISA and western blot.
Minor changes
- Page 1 lines 15-20, please rephrase this sentence. This is too long.
- What is the meaning of the lower activities of serum amylase in the combined treatment? Why is this not observed for serum lipase? Please explain the results in the main text or discussion.
- In figure 2E, what is the approach to obtain the GSH content? Please provide this information in the figure caption and experimental details in the method section.
- The western blot figure in figure 3, it is not clear for the MCP-1, ICAM-1 and TNF-α. Please provide a clearer result.
- In figure 4, what is the approach to obtain the concentrations of anti-oxidant indicators? Please provide this information in the figure caption and experimental details in the method section.
Author Response
Dear reviewer,
We have done the revision of our manuscript. I believe that this version is much better than the previous ones. I also believe that its scientific value is obvious and significant.
Thank you so much for all your help.
Sincerely,
Szu-Chuan Shen, corresponding author
Reply to comments of reviewer:
Major changes
- There is no introduction to the purpose of using pioglitazone and aminoguanidine in this study. Why are these selected for the study besides the VES?
Answer:
Aminoguanidine (AG) and Pioglitazone are different types of commercial hypoglycemic drugs and commonly used in many diabetes-associated studies. We compared the ameliorative effect of VES with these drugs, and further investigates the possible mechanism of VES on alleviating inflammation of pancreatic beta-cells in MG-treated rats in this study. The relevant descriptions of AG and Pioglitazone have been added in the text (lines 230-236).
- In the results section, the level of several groups of targets are learnt, including 1. Amylase and lipase; 2. RAGE and insulin secretion-related protein; 3. Inflammation proteins; 4. Antioxidant enzymes; and 5. Kinases. However, it was not explained in the study introduction why some of these proteins are selected, such as the selection of kinases. Please provide a more systematic introduction on how all of these types of proteins are selected as readout to evaluate the effect of VES.
Answer:
Accepted and revised accordingly. Previous study in our laboratory found the anti-inflammatory and anti-hyperglycemic effects of the polyphenol vescalagin (VES) in MG-induced carbohydrate metabolic disorder rats (Please see reference 20). This study further investigated the possible ameliorative mechanism of VES on pancreatic β-cells inflammation in MG-induced diabetic rats. The amylase and lipase, RAGE and insulin secretion-related protein, Inflammation proteins, antioxidant enzymes, and kinases were analyzed as indicators to elucidate the possible ameliorative mechanism of VES in MG-treated rats. The relevant statements were added in the manuscript (lines 91-93, 183-185).
- In all the figures, please describe the type of experiment conducted to obtain the corresponding figure. For example, what is the approach used to get figure 1? What is the readout of the experiment?
Answer:
Accepted and revised accordingly. The biochemical commercialize kits and ELISA reader were used for analyzing and calculating the experimental data of pancreatic inflammation indices and anti-oxidant indicators (Figure 1, Figure 4). The UVP Biospectrum image system and image J software were used for quantifying the results of western blotting (Figure 2, Figure 3, Figure 5). The relevant statements were added in the manuscript (lines 350-370, 377-395).
- For all the figures, what are the corresponding p-values of different letters a-c?
Answer:
The statistical data confirming that the significant difference between different letters is up to 99.5% (p < 0.05), and there is no check of statistical values between a-b, a-c or b-c in this study.
- Please provide the experimental methods for collecting pancreatic beta cells for ELISA and western blot.
Answer:
Accepted and revised accordingly. The relevant statements were added in the manuscript (lines 342-345).
Minor changes
- Page 1 lines 15-20, please rephrase this sentence. This is too long.
Answer:
Agree. The relevant sentences were revised accordingly (lines 15-20).
- What is the meaning of the lower activities of serum amylase in the combined treatment? Why is this not observed for serum lipase? Please explain the results in the main text or discussion.
Answer:
Accepted and revised accordingly. The relevant descriptions were revised in the text (lines 213-215, 222-224).
- In figure 2E, what is the approach to obtain the GSH content? Please provide this information in the figure caption and experimental details in the method section.
Answer:
Accepted and revised accordingly. GSH was analyzed by ELISA kit (Randox Laboratories, Crumlin, Antrim, UK) and performed follows the supplier’s protocols based on an enzymatic cycling method in the presence of GSH and a chromophore. The reduction of the chromophore produces a stable product, which can be followed kinetically at 450 nm. Therefore, its absorbance is directly proportional to the amount of GSH in the sample. The relevant descriptions were added in the manuscript (lines 350-357).
- The western blot figure in figure 3, it is not clear for the MCP-1, ICAM-1 and TNF-α. Please provide a clearer result.
Answer:
Agree and original Figure 3 was revised accordingly.
- In figure 4, what is the approach to obtain the concentrations of anti-oxidant indicators? Please provide this information in the figure caption and experimental details in the method section.
Answer:
Agree and revised accordingly. The approaches to obtain the concentrations of SOD, catalase and malondialdehyde (MDA) contents were added/revised in the caption of Figure 4 and Materials and Methods section (lines 357-370).
Reviewer 2 Report
In their manuscript, Chang et al. examined the effect of vescalagin intake on oxidation enzymes, inflammatory proteins and the insulin secretion-related proteins in pancreatic β-cells in rats. It is a very interesting topic and a well written manuscript.
Author Response
Dear reviewer, thank you very much for the great comment.
Reviewer 3 Report
This manuscript described the investigation of the occurrence of inflammation on pancreatic beta cells in MG-induced diabetic rats and the mechanism of ellagitannin vescalagin (VES) to prevent it. The authors found out that VES lowered the activity of amylase, downregulated the expressions of advanced glycation end products receptor, upregulated the protein expressions of pancreatic-duodenal homeobox-1 and downregulated CCAAT/enhancer binding protein-beta and elevated glutathione content, nuclear factor-erythroid 2-related factor 2, glyoxalases I and antioxidant enzymes, and then downregulated c-Jun N-terminal kinase and p38 mitogen-activated protein kinases pathways to protect pancreatic beta-cells in MG-administrated rats. A few suggestions:
- In the results session, the authors just presented the figures and briefly explained the figures. It would be easier for the readers to follow if there are some rationales presented about the reasons of doing these experiments. Although the authors did this in the discussion session, it would make more sense to rearrange some of the contents in the discussion session to the results session.
- In the results session, the authors compared VES with two other compounds PIO and AG. While the authors stated in the discussion session that PIO is a PPAR gamma ligand, no information was provided for AG. It would be reasonable to give some brief introduction about these compounds including their structures and mechanisms of action.
- From the results session, it looks like that VES is having similar activities as PIO and AG. While the authors wanted to be more focused on VES, more information or discussion about VES will be needed to differentiate this compound from the other two.
- From the title the authors seemed trying to investigate the anti-inflammatory activity of VES. However, they did not measure any of the cytokines except TNF-alpha.
- Some of the data from western blot are not very clear.
Author Response
Dear reviewer,
We have done the revision of our manuscript. I believe that this version is much better than the previous ones. I also believe that its scientific value is obvious and significant.
Thank you so much for all your help.
Sincerely,
Szu-Chuan Shen, corresponding author
Reply to comments of reviewer:
- In the results session, the authors just presented the figures and briefly explained the figures. It would be easier for the readers to follow if there are some rationales presented about the reasons of doing these experiments. Although the authors did this in the discussion session, it would make more sense to rearrange some of the contents in the discussion session to the results session.
Answer:
Agree and we revised/added more discussions in the Results and Discussion sections accordingly (lines 98-100, 116-119, 123-124, 138-139, 144-146, 166-170 and 183-185).
- In the results session, the authors compared VES with two other compounds PIO and AG. While the authors stated in the discussion session that PIO is a PPAR gamma ligand, no information was provided for AG. It would be reasonable to give some brief introduction about these compounds including their structures and mechanisms of action.
Answer:
Accepted and the relevant information of AG was added in the text accordingly (lines 233-236).
- From the results session, it looks like that VES is having similar activities as PIO and AG. While the authors wanted to be more focused on VES, more information or discussion about VES will be needed to differentiate this compound from the other two.
Answer:
AG is a nucleophilic hydrazine derivative and has been recognized as an antiglycation agent that reacts rapidly with aldehydes, such as MG, thereby preventing AGE formation. Pioglitazone is a peroxisome proliferator-activated receptor-γ (PPARγ) ligand, and an insulin sensitizer used for controlling the serum glucose of type 2 diabetes mellitus. So far, there are no studies investigates the anti-diabetic effect of VES, AG, and PIO in a single study. Thus, we first selected these two well-known commercial drugs as the positive control group to evaluate the anti-diabetic potential of VES in our study. The relevant descriptions were revised in the manuscript (lines 230-236).
- From the title the authors seemed trying to investigate the anti-inflammatory activity of VES. However, they did not measure any of the cytokines except TNF-alpha.
Answer:
Previous study in our laboratory found that VES significantly reduced serum IL-6 by 51.7%, in MG-induced diabetic rats (Please see reference 20). We further found that VES administration significantly reduced pancreatic inflammation-related proteins, such as NF-κB, ICAM-1 and TNF-α in MG-administered rats (Figures 3) in this study.
- Some of the data from western blot are not very clear.
Answer:
Accepted. The original Figure 3 was revised and redraw.
Round 2
Reviewer 1 Report
All the comments and questions are addressed in the answers and this is suggested to be accepted in the present form.
Author Response
Thank you very much for the great suggestion. A visiting professor from USA assisted us to improve English presentation.
Reviewer 3 Report
In this revised manuscript, the authors did some improvement. However, there is still something that need to be addressed.
- There is not enough rational provided about using PIO and AG as comparing agents. It is k that all three drugs have different mechanisms of action as hypoglycemic drugs, but they had very similar results in this study.
- In the result session, there is not enough explanation about the selection of the studied proteins and kinases. Also, not many differences have been observed among the three studies compounds.
- In the discussion session, the authors talked about resveratrol. Does Ves, as a polyphenol compound, act differently as compared to resveratrol? What are the differences between Ves and resveratrol in protecting pancreatic beta cells?
Author Response
1. There is not enough rational provided about using PIO and AG as comparing agents. It is that all three drugs have different mechanisms of action as hypoglycemic drugs, but they had very similar results in this study.
Answer: Accepted and revised. PIO and AG are currently clinical anti-diabetic drugs. The relevant description was revised in the text (lines 231-236). This study compared the pancreatic inflammation indices, inflammation protein expressions, RAGE and insulin secretion-related protein expressions and antioxidant enzymes activities of these two compounds as well as VES in MG-rats. Although some results of these three compounds are very similar, however, it is interesting that some observed founding of these three compounds are different (ex. Amylase, RAGE, GSH, pNrf-2/Nrf-2 etc.) in the current study, indicating that hypoglycemic mechanisms of them may be not all the same.
2. In the result session, there is not enough explanation about the selection of the studied proteins and kinases. Also, not many differences have been observed among the three studies compounds.
Answer: Agree and revised. Previous study in our laboratory found that serum inflammatory TNF-α and IL-6 contents in the MG group were significantly higher than those in normal group (lines 219-221, reference 20). Therefore, we further focus on investigating NFkB-related inflammation proteins and kinases to evaluation the anti-inflammation capacity/mechanism and found some results of these three studied compounds on the dosages used (30mg/kg BW) in this study are not significantly different compared to each other, indicating the exact hypoglycemic mechanisms of these three compounds may be something different.
3. In the discussion session, the authors talked about resveratrol. Does Ves, as a polyphenol compound, act differently as compared to resveratrol? What are the differences between Ves and resveratrol in protecting pancreatic beta cells?
Answer: Thank you very much for the good suggestion. Since the evidences on protective effects/mechanisms of resveratrol are insufficient in MG-rats, the relevant description has been corrected in the text (lines 222-224).